# The Effects of Galactic Cosmic Rays on the Central Nervous System: From Negative to Unexpectedly Positive Effects That Astronauts May Encounter

**DOI:** 10.3390/biology12030400

**Published:** 2023-03-03

**Authors:** Viktor S. Kokhan, Mikhail I. Dobynde

**Affiliations:** 1V.P. Serbsky Federal Medical Research Centre for Psychiatry and Narcology, 119034 Moscow, Russia; 2Skobeltsyn Institute of Nuclear Physics, Moscow State University, 125009 Moscow, Russia

**Keywords:** manned interplanetary mission, ionizing radiation, galactic cosmic rays, cognition, emotional state, neurogenesis, neurodegeneration

## Abstract

**Simple Summary:**

The central nervous system is extremely sensitive to cosmic rays – an ionizing radiation that astronauts encounter during interplanetary missions, particularly to Mars. There are still disputes about how dangerous cosmic rays are to brain health. Our review aimed to analyze the studies on the influence of cosmic rays at flight-related doses on physical development, well-being, emotional state, depressive-like behavior, cognition, neurogenesis and the course of neurodegenerative process. The combined effects of cosmic rays and hypogravity were also the focus of our review. Thus, for the first time a comprehensive picture of radiation-induced changes in the central nervous system is presented: from functional to neurochemical and molecular. Relying on the literature data we conclude that cosmic rays are relatively safe for the central nervous system functions. Moreover, under some irradiation scenarios, it has been shown that cosmic rays enhance cognitive abilities of rodents and nonhuman primates. This effect is often accompanied by hyperactivity, increased orientational-exploratory behavior and, simultaneously, state anxiety. Interestingly, the results of several studies strongly suggest that cosmic rays reverse negative effects of antiorthostatic suspension (a ground-based model of hypogravity). Taken together, these data allow us to be optimistic about the prospects for humanity’s cosmic expansion.

**Abstract:**

Galactic cosmic rays (GCR) pose a serious threat to astronauts’ health during deep space missions. The possible functional alterations of the central nervous system (CNS) under GCR exposure can be critical for mission success. Despite the obvious negative effects of ionizing radiation, a number of neutral or even positive effects of GCR irradiation on CNS functions were revealed in ground-based experiments with rodents and primates. This review is focused on the GCR exposure effects on emotional state and cognition, emphasizing positive effects and their potential mechanisms. We integrate these data with GCR effects on adult neurogenesis and pathological protein aggregation, forming a complete picture. We conclude that GCR exposure causes multidirectional effects on cognition, which may be associated with emotional state alterations. However, the irradiation in space-related doses either has no effect or has performance enhancing effects in solving high-level cognition tasks and tasks with a high level of motivation. We suppose the model of neurotransmission changes after irradiation, although the molecular mechanisms of this phenomenon are not fully understood.

## 1. Introduction

After the success of the Apollo 11 mission, the cosmic ambitions of humanity have been limited to the Earth’s orbit and launching automatic probes to outer space. The global scientific and technological progress and the discovery of water on the Moon and Mars surface [1,2,3] made the return of man to the Moon possible, as well as manned flights to other planets (primarily to Mars) with the planned long-term work on the surface. The inevitability of building habitable Martian and Lunar bases becomes obvious today.

It is well-known that the space flight factors (SFF), such as ionizing radiation (IR), hypomagnetic field, hypo- and hypergravity, isolation, changes in the gas environment, etc., have a significant impact on living organisms. Orbital flights are a good source of the data for all factors, except IR and hypomagnetic field [4]. It is due to the protective effect of the Earth—the magnetic field modifies the GCR spectra in the “low” energy range, while the Earth body reduces the GCR intensity in the whole energy range. Beyond the geomagnetic field (the magnetopause starts ~75,000 km from the surface), the IR intensity increases due to a flux of heavy charged particles (HZE)—the most dangerous component of galactic cosmic rays (GCR) whose sources are outside the Solar System. The comparative analysis of different IR in relation to living organisms indicates an extremely high biological effect of HZE due to the high ionization density along their tracks [5]. Emphasis on the long-term effects of radiation, such as oncogenesis, cataractogenesis, retinal dystrophy, and bone marrow dysfunction, researchers left the CNS response to radiation damage outside the view until the long interplanetary mission’s task was set. Indeed, immediate CNS functional disruptions caused by HZE become critically important for mental health and creative abilities for piloting a spaceship and managing life support during the mission. Thus, the delayed stochastic irradiation effects remain in focus but are not limiting in comparison with immediate CNS functional disruptions [4,6,7].

The study of the GCR effect on living organisms is complicated by its multicomponent nature, chronic low-dose exposure, induced radioactivity, bioethical issues while working with humans, and the mismatch of radiation doses and composition on Earth (accidents, nuclear weapons using consequences, and therapeutic radiotherapy). Thus, the data from astronauts at the International Space Station (ISS), orbital and ground modeling tests using primates and rodent models, remain the main data source on the effect of GCR on CNS.

The last decade’s studies on the GCR effects have significantly intensified, in particular regarding CNS functions. Most reviews are focused on the negative effects of GCR on the cognitive abilities and emotional state (fear, anxiety, arousal, emotional reactivity (ER), etc.) of laboratory animals [6,8]. At the same time, there is a lot of recent data indicating pro-cognitive [9,10,11,12], antidepressant [13,14], and nootropic [15,16] effects when laboratory animals were exposed to IR of various doses and composition, including space-related. Such effects in relation to CNS can be classified as positive, especially taking into account their persistent nature [9]. For small doses of electromagnetic IR, we can explain this phenomenon by radiation hormesis that is even used in physiotherapeutic practice [17], but it remains poorly understood which mechanisms underlie the positive effect of neutrons and corpuscular IR, including H^+^ and heavier nuclei in space-related and even higher doses.

This review is devoted to the analysis of the literature on the effect of space-related IR doses on CNS with an emphasis on its positive effect and its possible mechanisms.

## 2. Galactic Cosmic Rays

A number of works cover the issue of IR composition and doses both in outer space and on the Moon and Mars surfaces [18,19,20,21,22]. Here we are considering this issue insofar as it is necessary for the classification and interpretation of biological data.

GCR contain 2% of electrons and 98% of atomic nuclei, which consist of about ~87% H^+^, 12% ^4^He, and 1–2% HZE—are very deeply ionizing highly charged nuclei with an atomic number (Z) more than two (Figure 1) [23]. The galactic flux of neutrons (~11 mSv/year) and γ-rays is low; therefore, the main source of IR on-board of spaceship is H^+^, HZE, and delta rays caused by the interaction of HZE with spacecraft hull, equipment, and living tissues [24,25,26]. An important HZE parameter is linear energy transfer (LET)—the rate of particle energy loss, measured in keV/µm of water. The vast majority of collisions will be produced by low-LET particles, while the number of collisions and, accordingly, the absorbed dose for high-LET particles, will be relatively small (Figure 2). At the same time, the biological effect of space radiation cannot be characterized only by the absorbed dose, which is measured in units of gray (Gy). Instead, the dose equivalent measured in the units of sievert (Sv) is widely used. When calculating the equivalent dose, we should take into account the IR quality factor (Q: tissue’s track structure concept; function of LET and Z). Thus, the equivalent dose makes it possible to correctly compare irradiation by nuclei with different values of Z and energy, as well as combined irradiation. However, the dose alone is not an exhaustive predictor of biological effectiveness; particle fluence is also important since it determines the number of affected cells [27,28].

The flux of GCR particles in interplanetary space fluctuates inversely with the solar cycle, from doses of 50–100 mGy/year at solar maximum to 150–300 mGy/year at solar minimum [18]. An average total GCR dose rate at Gale Crater on Mars is 0.21 ± 0.04 mGy/day compared to 0.48 ± 0.08 mGy/day in interplanetary space. The average Q on the Martian surface is 3.05 ± 0.3 compared to 3.82 ± 0.3 in interplanetary space [20]. The authors estimate the equivalent dose that astronauts will receive during the 860-day Martian mission (360 days of both directional flight and 500 days on the surface) as ~1 Sv.

The GCR dose for astronauts on ISS is estimated as 38–190 mSv/year and depends on the orbit and solar activity [34]. However, to obtain the total equivalent dose, the contribution of electrons and H^+^ trapped in the Van Allen radiation belt must be added. The inner belt is closest to the Earth in a region of the Brazilian coast called the South Atlantic Anomaly, which extends from about 0° to 60° W and 20° to 50° S (geographic coordinates) and is the ISS flight zone [35]. Taking this into account, the total dose equivalent received by the astronauts at ISS is 167–295 mSv/year with a Q value between 2.6 and 2.9 depending on the solar activity and the measurement point inside the station (because of the shielding by external elements of the station hull) [36]. These data indicate that orbital flights are close to the IR load model on the Martian surface but not during the flight in an interplanetary space. Indeed, despite the lower dose of HZE onboard ISS, the composition (Figure 1) and the chronic nature of the exposure makes orbital flight an actual model of radiation load on the Mars surface. Moreover, on the surface of Mars, astronauts will be additionally protected by the hull of the living module, which will lead to some reduction in the equivalent dose.

Unpredictable solar particle events also should be taken into account when calculating the final dose. However, their effect is not great—most of solar origin protons have energy lower than ~100 MeV, so they could be efficiently stopped with shielding. At the same time, astronauts onboard an interplanetary ship will be better protected than the dosimetry equipment from which the data was obtained previously. Indeed, using the protective sheathing made of polyethylene with a density of 30 g/cm^2^ can reduce the equivalent dose by 30% [18]. Thus, the dose relevant to Martian (deep space in general) mission cannot be determined precisely; we suggest focusing on an approximate range of 0.5–2 Sv. The use of a range instead of the exact value will allow taking into account the higher (in ~1.7 time) radioresistance of rodents [4]—the most common animal model in ground-based experiments.

Modeling the IR environment of interplanetary space and/or the surfaces of other planets under ground-based conditions faces a number of almost insurmountable obstacles: the multicomponent composition of elementary particles and nuclei, chronic low-dose exposure, unpredictability of particle tracks, etc. A lot of early studies used a specific ion in a monoenergetic beam for exposure. These studies are important to understand the specific mechanism of the radiation damage formation but do not reproduce the real radiation environment of interplanetary space. More advanced models assume a mixed irradiation of protons and HZE or γ-rays and HZE, and the HZE irradiation can be fractionated, which reduces the acute dose [10,37,38]. The use of γ-rays as part of the combined irradiation allows prolonged irradiation, as well as the study of the combined effect of factors, such as IR and hypogravity (antiorthostatic suspension model) [39]. It was found that in some cases with sequential irradiation the effect depended on the beam order [40], in some cases did not [41], and in some cases sequential beam exposure produced a different effect comparing to simultaneous exposure [42]. Another approach is to use a specially designed target to produce a field of secondary particles [43]. There is an option to use the radioisotope neutron sources for modeling chronic low-dose exposure, for example ^252^Cf [44], but the neutron flux only reproduces the high-LET component of GCR [45]. Thus, even the most progressive models are not entirely perfect, which must be kept in mind when interpreting the results.

## 3. Well-Being, Weight, Locomotor Abilities, and IR

Few studies have been devoted to the effect of IR on lifespan. Acute irradiation by γ-rays (~1.3 MeV) leads to a shortened lifespan in female B6CF1 mice at 0.9 Gy but not 0.23 or 0.45 Gy [46]. In contrast, chronic γ-rays irradiation (^232^Th source; 70 or 140 mGy/year) at an early age (3–4-week-old) results in a ~23% increase in lifespan of female C57BL/6 mice at a time point corresponding to 50% survival time of the population [47]. At the same time, in a nest building test, chronic irradiation (^252^Cf source, 1 mGy/day; 0.12, 0.2, or 0.4 Gy totally) was shown to improve the well-being of female mice 18 months later after the irradiation. In the same experiment, the authors showed that irradiation only at an absorbed dose of 0.2 Gy led to a statistically significant increase in body weight in male mice 21 months after the irradiation, but in absolute terms this difference is insignificant [12]. Other studies have found a decrease in the body weight of rats in 7 months after irradiation (0.4 Gy γ-rays and ^12^C 0.14 Gy, 10.3 keV/μm) [9]. A number of studies revealed no effect of irradiation on the body weight of mice under different irradiation scenarios: ^28^Si (0.2 or 1 Gy, 67 keV/μm), ^56^Fe (0.1 or 0.5 Gy, 151.4 keV/μm), and mixed HZE (H^+^, ^4^He, ^12^C, ^16^O, ^28^Si, ^48^Ti, ^56^Fe with different energy, 0.75 Gy totally) [16,37,48]. Finally, ^12^C irradiation (50 or 100 mGy, 15 keV/μm) had no effect after ~12 and ~24 months on the body weight, total brain mass, and hippocampal mass of the mice irradiated at 10 weeks [49]. Thus, in spite of the effects detected, IR has no critical effect on the physical development of rodents.

It is important to establish the IR effect on locomotor abilities, as most behavioral tests work only if the animal moves. The irradiation by low-LET H^+^ in large doses (3 or 4 Gy, 0.39 keV/µm) leads to impaired performance in the rotarod test [50]. At the same time, the irradiation by ^12^C nuclei in different scenarios (2 or 5 Gy, 10–20 keV/µm; 2 Gy, 70–100 keV/µm) did not change rotarod test performance or average swimming speed in the Morris water maze [51]. More important, there are no changes in the rotarod test performance in space-related 0.1 Gy and higher (0.5 or 2 Gy) doses of ^56^Fe (148.2 keV/µm) observed [52].

It was shown that ^56^Fe (0.5 Gy, 175 keV/µm) irradiation of mice leads to a motor activity decrease [53]. High-LET H^+^ irradiation (1.5 Gy in spread-out Bragg peak: <25 keV/µm—67%, 25–50 keV/µm—23%, and 50–100 keV/µm—10%) in combination with γ-rays (3 Gy) does not affect the motor activity [14]. In contrast, the combined low-LET irradiation causes hyper-locomotion: H^+^ (1.5 Gy, 0.4 keV/µm) and 3 Gy γ-rays; 0.14 Gy ^12^C (10.3 keV/μm), and 0.4 Gy γ-rays [39,54]. In the more relevant GCR model—mixed radiation by H^+^ (0.24 keV/μm, 60% in absorbed dose), ^16^O (25 keV/μm, 20%), and ^28^Si (78 keV/μm, 20%)—an increase in locomotor activity was found only at a total absorbed dose 0.5 Gy but not at 0.25 Gy or 2 Gy [55]. Ultimately, none of the studies identified critical abnormalities that could distort the behavioral tests results in space-related doses.

We believe, the impairment of monoaminergic and, first of all, dopaminergic neurotransmission is responsible for performance impairments in the rotarod test at high (not relevant to outer space) doses [56]. *Inter alia*, the decrease in motor activity may be associated with radiation-induced neuroinflammation [57,58,59]. The mechanisms of the stimulating effect of irradiation on motor activity remain poorly understood. One possible mechanism could be a radiation-induced decrease of dopamine content in the nucleus accumbens (by ~50%) [60], which, based on the study results [61], is directly associated with hyperactivity.

## 4. Mental Health and Ionizing Radiation

The astronaut’s mental health depends on comfortable working conditions, atmosphere, social and workplace interaction, and ultimately plays a paramount role in the mission’s success [62,63,64]. Significant changes in the psycho-emotional state (emotions of negative and positive valence) under SFF influence were detected both in astronauts during orbital missions and in laboratory animals involved in ground-based experiments [4,65]. An increased fatigue, impaired attention, sleep disturbances, and transient anxiety [66,67] were described in astronauts, but these states are not obligatory. In one of the latest NASA experiments “Twins study”, the astronaut did not experience any critical psycho-emotional disturbances during the 340-day stay at ISS [68].

At present, most of the data on the IR effect on the emotional state has been obtained during ground-based modeling using laboratory rodents, where the radiation effect alone can be distinguished. Early studies showed that IR acts as a moderate unconditioned aversive stimuli for animals [69]. Importantly, animal avoidance of irradiation zones is not associated with the phenomenon of phosphenes, pituitary and adrenal gland functions, as well as pain [70,71]. Subsequent works redefined this formulation by pointing to the drive (defined by autonomic nervous system status) level enhancing effect of IR [72], which is basically determined by ER. Later, this phenomenon was designated as disinhibition of the CNS [4].

### 4.1. Anxiety and Ionizing Radiation

One of the main components of emotional states is the level of anxiety. The vast majority of studies assess the level of “state” anxiety (hereinafter anxiety) under the conditions of novelty or the action of stressful factors. At the same time, the IR effects remain poorly understood for the “trait” anxiety (innate characteristic).

The large doses of IR do not affect the level of anxiety—no anxiety alterations were detected when rats irradiated by ^56^Fe (1.5 Gy, 147 keV/µm) or combined 3 Gy γ-rays and 1.5 Gy H^+^ (0.4 keV/µm) [39,73]. However, γ-irradiation alone in the dose range of 0.5–2 Gy leads to an anxiety increase [74]. Irradiation by H^+^ and HZE in space-related doses has a similar effect. The irradiation by H^+^ increased anxiety at doses of 0.5 or 1 Gy (0.5 keV/µm) [75]. The irradiation by heavier nuclei also caused anxiety increase: ^4^He (1–100 mGy, 0.9 keV/µm), ^28^Si (1 Gy, 67 keV/µm), or ^48^Ti (0.3 Gy, 126 keV/µm) [48,58,74]. It is noteworthy that aging animals are more sensitive to the anxiogenic effect of HZE: 15-month old rats exhibit anxious behavior during irradiation by ^48^Ti (10 or 100 mGy, 134 keV/µm) and ^16^O (1 mGy, 14.2 keV/µm) but not when irradiated by ^4^He (0.1–1 mGy, 0.9 keV/µm), whereas rats (2 and 11 months old) do not show changes in anxiety under such doses [76]. Combined exposure (0.4 Gy γ-rays and ^12^C 0.14 Gy, 10.3 keV/μm) also has an anxiogenic effect [9,54]. Recently, the anxiogenic effect was shown for chronic exposure by neutrons and γ-rays (^252^Cf source, 1 mGy/day, 0.4 Gy totally). The chronic nature of the exposure makes this model extremely relevant to space flight; however, it should be taken into account that neutrons simulate exclusively the high-LET nuclei effect. At lower absorbed doses (~0.12 or 0.2 Gy) in the same exposure scenario, no effect on anxiety-related behavior was found [12]. In the study with the most progressive irradiation model (0.75 Gy, a combination of 33 nuclei) there was no change in motor activity and anxious behavior, but there was a detection of sociability deficits [37]. These data are summarized in Table 1. Thus, HZE in space-related doses predominantly has a neutral or anxiogenic effect; at the same time, the dose is of paramount importance.

It is known that both acute and chronic stress cause anxiety to increase in rodents. Therefore, moderate (space-related) IR doses can be considered as a stress factor. Ultimately, stress is the organism’s physiological response to harmful or threatening stimuli that allow appropriate behavioral responses to the stressor. If the organism could not adapt to stress, adaptive physiological responses would be converted into the maladaptive pathological conditions [78]. Several interconnected brain areas are involved in the control of stress and anxiety, especially the locus coeruleus, amygdala, and hypothalamic-pituitary-adrenal (HPA) axis are marked as the key components [79]. Indeed, the neurochemical picture of irradiated animals indicates an amygdala reactivity increase and HPA axis activation with space-related [54], but not large doses, of IR [14,39]. Interestingly, the anxiety behavior markers in the dark-light box (but not in the elevated plus maze) are known to positively correlate with the level of hippocampal neurogenesis [80,81]. Indeed, a discrepancy in the level of anxiety in these tests was found 7 months after IR (0.4 Gy γ-rays and ^12^C 0.14 Gy, 10.3 keV/μm) exposure [9]. These data indirectly indicate a strong link between the IR anxiogenic effect and the level of neurogenesis (see Section 8).

To date, little attention is paid to the study of molecular mechanisms of the GCR anxiogenic effects. However, several biomolecules, which may be involved in the irradiation anxiogenic effect, have been proposed. It is suggested that increased neurokinin-1 receptors expression in the amygdala and decreased NR1 type of N-methyl-D-aspartate receptors and GABA transporter 1 expression in the neocortex within 1 month after irradiation may be responsible for the IR-induced anxiogenic effect in young rats (3.5-month-old at analysis point) [9,54]. Recently, the involvement of the neurokinin-1 receptor in the IR-induced anxiogenic effect has been confirmed pharmacologically. At the same time, downregulation of 5-HT_2c_ serotonin receptor expression in the amygdala as well 5-HT_4_ in the hypothalamus can be considered as a neuro-adaptive anxiolytically-orientated response of nerve tissue [82]. Thereby, due to the anti-anxiety molecular changes, the anxiogenic effect of irradiation is probably not persistent. Indeed, the anxiogenic acute exposure effect does not last long [83,84]. In any case, the anxiogenic IR effect phenomenon requires precise future research.

### 4.2. Depressive-like Behavior and IR

The IR effect on the depressive-like behavior of rodents is relatively poorly studied, but the available data is very intriguing. Irradiation by H^+^ (1.5 Gy, 0.4 keV/µm) did not affect the depressive behavior marker—freezing—in the open field test, but 3 Gy led to a significant increase of depressive-like behavior markers [85]. Combined irradiation (H^+^ and HZE) causes depressive-like behavior (forced swim test) at the 0.5 Gy dose, while in the other studied doses (0.25, 2 Gy) no significant effect was found [55]. At the same time, multiple scenarios of combined exposure (γ-rays 3 Gy and H^+^ 1.5 Gy, spread-out Bragg peak; γ-rays 3 Gy and H^+^ 1.5 Gy, 0.4 keV/µm; γ-rays 0.4 Gy and ^12^C 0.14 Gy, 10.3 keV/µm) had an antidepressant effect, which is to reduce freezing episodes and time of freezing in the open field test in rats [14,39,54]. Mixed fields irradiation (^252^Cf source: 0.8 mGy/day neutron, 0–15 MeV, 0.2 mGy/day γ-rays, chronically for 180 days) does not cause depressive-like behavior (forced swim test) in mice [44]. Interestingly, irradiation with γ-rays 3 Gy (fractionated 0.5 Gy × 6, twice per day) blocks molecular changes associated with chronic benzodiazepine treatment and related to depressive-like behavior [13]. These data may indicate that electromagnetic IR affects depressive-like behavior more than IR corpuscular nature—suppression of depressive-like behavior markers occurs under γ-rays and combination γ-rays and H^+^ or HZE irradiation.

The molecular mechanisms of IR antidepressant effect are still not disclosed. The alteration of several receptors’ expression has been suggested as possible pathways: dopamine D_2_ increasing in the striatum and serotonin 5-HT_2a_ and 5-HT_3_ decreasing in the hypothalamus and the amygdala [39,54].

Despite the fact that freezing in rodents can be considered as a marker of depressive-like behavior [86]; this marker remains only an indirect sign. Ultimately, space-related irradiation does not induce depressive-like behavior, and the data on the antidepressant effects requires validation. Further research using targeted behavioral tests (Porsolt or tail suspension for mice), imipramine-validated animal models of depressive-like behavior, and primates with clinical depression is required.

### 4.3. Protons and HZE Stimulate Habituation, Orientation and Exploratory Behavior

Moderate doses of H^+^ irradiation stimulate orientation and exploratory behavior (OEB): 1 Gy H^+^ 0.4 keV/µm or 1.5 Gy H^+^ (spread-out Bragg peak) [14,87]. Moreover, this effect can be stable up to 90 days after irradiation [87]. It is noteworthy that in irradiated rodents (^56^Fe, 0.1, 0.5, 1 or 2 Gy, 148 keV/µm), a new object placed in an open field caused increased interest [52,73]. On the contrary, the combined exposure (0.14 Gy ^12^C and 0.4 Gy γ-rays) did not cause changes in OEB [54], while large doses of low-LET particles (H^+^, 3 Gy, 0.4 keV/µm) suppressed OEB [85]. Thus, we observe a certain effective fluence and LET range, outside which the effect is either not detected or reversed.

It is generally accepted that OEB and anxious behavior (fear) are connected by a negative correlation [88]. However, the previously observed data do not indicate any connection between OEB activation and total motor activity, anxiety, or depressive-like behavior markers of irradiated animals. It must be emphasized that OEB often is considered as a reaction mediating undirected attention and serves as an indirect predictor of general learning factor [89].

In the forced novelty conditions, OEB is strongly connected to a risk-taking tendency and ER. For γ-rays (3 Gy), H^+^ (1.5 or 3 Gy, 0.4 keV/µm), and combined IR (^12^C, 0.14 Gy, 10.3 keV/µm and 0.4 Gy γ-rays) exposure neither ER nor risk-taking behavior were found [54,85,90]. At the same time, H^+^ exposure at a lower dose (1 Gy, 0.4 keV/µm) enhances ER [87].

Several works have revealed the increased habituation in response to irradiation both in a novel and familiar environment: 3 Gy γ-rays; 1 Gy H^+^ (0.4 keV/µm) after 30, but not after 90 days, 1.5 Gy or 3 Gy H^+^, as well as combined IR (0.14 Gy ^12^C and 0.4 Gy γ-rays; after 1, but not after 212 days) [9,85,87,90]. On the one hand, rapid habituation correlates positively with OEB [91,92] and adaptive, nonpathological anxiety [93], which fully corresponds to the emotional status of animals irradiated at space-related doses. The direction of this correlation is also confirmed in the study, which showed the absence of irradiation effects on habituation against the background of anxiolytic therapy [82]. On the other hand, in both humans and animals, the disruption of habituation is strongly correlated with cognitive impairments [94,95] and vice versa [93,96]. Thus, both active OEB and rapid habituation of irradiated rodents indirectly indicate a possible enhancement of cognitive abilities.

To date, there is almost no data on the possible mechanisms responsible for changes in OEB, ER, and habituation upon what HZE irradiation. It has been suggested that a decrease in GABA innervation in the hippocampus and adjacent structures may be responsible for the rapid habituation [54].

## 5. Cognition and Ionizing Radiation

A number of reviews provide a complete picture of the IR nature, dose, LET, Z, and fluence (in case of corpuscular IR) effect on cognitive abilities (rodents) and operator activity elements (primates) [4,6,7,8]. Of particular note is the work presenting integrated information about negative/neutral IR effect in various behavioral paradigms (dose, LET, Z of GCR corpuscular component and time after irradiation) [7]. At the same time, the positive effects of H^+^ and HZE irradiation are ignored, and the neutral effects are not discussed.

To date, there is no established theory to explain the multidirectional effects of IR (including HZE and combined IR) on different memory types and cognitive task performance. Different authors often describe contradictive data at the same radiation value and animal species. Probably, there are many more studies that might show the neutral or even positive effect of IR, but publishing such results is associated with difficulties. Other scientists also share this point of view [97].

### 5.1. Primate Studies

According to the experiments with *Macaca mulatta*, irradiated (acute 3.5 Gy or fractionated: 1 Gy × 10, 14 days interval; 2 Gy × 10 X-ray, 200 keV) animals showed a tendency to more efficient complex problem solving [98]. It is noteworthy that ultralow doses of natural combined exposure, including GCR (carried out at an altitude of ~28 km above the Earth’s surface) ~0.216 mGy with Q = 4 (our calculation), also led to the cognitive improvement of *Macaca fascicularis* [99]. An irradiation of *Macaca mulatta* by H^+^ (3 Gy, 0.4 keV/µm, head only) leads to the improved learning performance in operator’s activity elements simulator (in particular, the tracking tasks) that requires interaction with a computer (joystick). Thus, the very high fluence of low-LET H^+^ can enhance the performance in high-level cognition task. After 40 days, the same monkeys were re-irradiated by ^12^C (1 Gy, 10 keV/µm, head only), which caused learning disabilities in monkeys with the strong and unbalanced (excitable) but not with the strong and balanced type of higher nervous activity [11]. Recently, these data have been confirmed using lower doses of IR (combined 1 Gy γ-rays, 0.23 Gy/h and ^12^C, 1 Gy, 10 keV/µm), but it should be kept in mind that in this study the combined effects of IR and antiorthostatic hypokinesia were used [100]. Taken together, this data confirms the paramount role of the animals’ emotional state in the manifestation of HZE effects on CNS functions, which was hypothesized previously [54].

### 5.2. Fear and Contextual Memory, High-Level Cognitive Tasks

Interestingly, the first data on the positive IR effect on cognitive functions were obtained in the ’50s of the last century: training rats in the T-maze improves after 5 Gy X-ray irradiation [101]. At the same time, rodents acquire several cognitive benefits after being exposed to H^+^ and HZE. The first studies using low-level behavioral paradigms have shown that the exposure to ^56^Fe (1 or 2 Gy) increased the freeze response of male but not female mice in the contextual fear conditioning test. At the same time, in the cued fear conditioning test, no IR effects (1–3 Gy) were found [102]. A fluence increase and LET decrease (^16^O) taken together lead to contextual fear memory improvement (0.4, 0.8, but not 1.6 Gy, 25 keV/μm) but have no effect on cued fear memory [103]. Irradiation by H^+^ (0.1 Gy, 0.5 keV/μm) also enhance contextual fear memory [104].

Rats exposed to H^+^ (1.5 Gy, 0.4 keV/μm) demonstrated a faster and more effective (half the failure rate) learning in Y-maze with sound as the conditional stimulus and electric foot-shock as the unconditional stimulus [85]. Long-term contextual memory improves and short-term memory stays unaffected in the passive avoidance test (PA) after H^+^ (2, but not 1 Gy, spread-out Bragg peak) [105]. The study with the more progressive model of irradiation in space-related equivalent dose of ~2 Sv (our calculation from 0.25 Gy), which consisted of H^+^ (0.15 Gy, 0.24 keV/μm), ^16^O (50 mGy, 25 keV/μm), and ^28^Si (50 mGy, 78 keV/μm) and even larger doses of mixed nuclei in the same proportion (0.5 or 2 Gy, totally) does not detect fear conditioning memory impairment in PA [55]. Several other studies also do not reveal abnormalities in PA after both alone and combined γ-rays, H^+^, and HZE irradiation [14,39,105]. The analysis of the data indicates that these last stimulating/neutral effects are not associated with changes in home cage activity and depressive-like behavior [55], open field motor activity, or anxiety [14,39]. It is noteworthy that the PA learning effectiveness shows one of the highest correlation values with the general learning factor of rodents [106]. An impairment in attentional set shifting task was identified after ^28^Si (0.05–0.2 Gy, 54 keV/μm) exposure. However, in some stage of the test, compound discrimination reversal, in particular irradiated rats (0.15 Gy), showed a better performance than naïve. The authors note a great variability in the experimental groups for radioresistance [28]. Recently, the high-level cognitive tasks with operator activity elements after the whole-body irradiation were studied on mice in 4 versions of irradiation: ^56^Fe (67 mGy × 3 or non-fractionated 0.2 Gy, 174 keV/μm) or ^28^Si (0.2 or 1 Gy, 72 keV/μm). There were no task performance violations; moreover, the irradiation leads to improved performance on a dentate gyrus-reliant pattern separation task [10].

### 5.3. Spatial Memory and Learning

Irradiation by H^+^ (1.5–4 Gy, 0.3 keV/μm) does not violate spatial learning in the Morris water maze (MWM) [107]; some alterations were detected only in the spatial memory retention (the probe test) [108]. More space-related combined irradiation (0.4 Gy γ-rays and ^12^C, 0.14 Gy, 10.3 keV/µm) enhanced spatial learning. Moreover, this effect was detected immediately after the irradiation and after 7 months [9,54]. The further LET increase and fluence decrease (^56^Fe, 0.1, 0.2 or 0.5 Gy, 175 keV/µm) does not affect spatial learning [109]. Moreover, 0.5 Gy ^56^Fe irradiation enhanced the spatial memory retention but did not have an effect on MWM spatial learning in transgenic mice (apolipoprotein E3 gene was replaced with human’s homolog) [110]. Interestingly, this is in accordance with the threshold fluence for neurogenesis disorders (see Section 8). Irradiation by ^16^O (0.05, 0.1, or 0.25 Gy, 15.8 keV/μm) did not affect the spatial memory in Y-maze right after the experiment and within the following several months [111,112]. However, short-term period disturbances were detected after the exposure to doses of 0.1 or 0.25 Gy but not 1 Gy [113]. At the same time, the spatial memory impairment was detected after ^56^Fe (0.2–0.6 Gy, 147 keV/µm) irradiation in Barnes maze [114]. Subsequently, a negative effect of 0.2 Gy was confirmed, but it was detected in several animals indicating radioresistance differences between individuals in the population. At the same time, no violations were detected while using a dose closer to space missions (50 mGy) [115]. Fluence increase and LET decrease (^48^Ti, 50 mGy, 106 keV/µm), on the contrary, lead to the spatial memory impairment [116]. This assumption was confirmed in another study: irradiation by ^4^He (1, 5, 10, and 50 but not 100 mGy) led to the memory impairment in object location recognition test, but these changes were associated with high anxiety [74]. In contrast, mixed HZE irradiation did not impair performance in this test at a space-related 0.25 Gy (H^+^, 0.15 Gy, 0.24 keV/μm and ^16^O, 50 mGy, 25 keV/μm, and ^28^Si, 50 mGy, 78 keV/μm) dose, whereas the impairment was observed at higher doses (in the same proportion of nuclei)—0.5 and 2 Gy [55].

### 5.4. Recognition Memory

Very intriguing and contradictory data were obtained in the analysis of the recognition memory for the object recognition test (ORT). It was shown that combined irradiation (^252^Cf source, 1 mGy/day: 0.8 mGy neutron, 0–15 MeV and 0.2 mGy γ-rays, chronically 180 days) led to impairments in a number of cognition tasks, including ORT. At the same time, the researchers noted a high level of anxiety [44]. However, in another study, with very similar parameters of chronic exposure (^252^Cf, 1 mGy/day, absorbed dose ~0.12 or 0.4 Gy), violations in the ORT test were not detected. Moreover, at the dose 0.2 Gy, both naïve and irradiated mice showed no interest to the new object, which indicates the questionable effectiveness of ORT in memory assessment. At the same time, the authors note improved nest building at 0.12, 0.2, or 0.4 Gy and confirm the anxiogenic effect at 0.4 Gy [12]. Significantly, this model reproduces only the high-LET component of HZE, and its daily equivalent dose is exceeded by ~10 times (~12 mSv/day, our calculation) compared to outer space. It was also noted by other authors [45]. Irradiation by ^56^Fe (0.25 Gy, 175 keV/µm) did not affect behavior in ORT [117]. Another study confirmed this data [118], but in doses of 0.1 or 0.4 Gy violations were detected. According to author’s hypothesis, cognitive changes are observed when the synapse remodeling response is either not initiated (0.1 Gy) or cannot cope (0.4 Gy) with radiation damage. However, this hypothesis is not confirmed when using significantly higher doses without violations. Thus, acute irradiation by ^56^Fe impairer memory of males only at the dose of 2 Gy (but not 1 or 3 Gy), while in females this dose was 3 Gy (but not 1 or 2 Gy), and this effect had no correlations with motor activity and anxiety [102]. Irradiation by ^16^O (0.3 Gy, 15.8 keV/μm) causes the recognition memory impairment, but these impairments are associated with high anxiety [58]. However, in other studies, authors showed the anxiety-independent memory impairment for ORT after ^4^He (0.15, 0.50, but not 1 Gy, 1.57 keV/µm) irradiation [84]. Recognition memory impairment was also detected after irradiation by ^16^O (50 mGy, 15.8 keV/μm) [111,112]. However, after irradiation with the same dose—50 mGy ^48^Ti (106 keV/µm), no disturbances were detected [58]. Low-LET irradiation in ultra-low doses (^4^He 1–50 mGy, 0.89 keV/µm) also does not impair recognition memory [74]. Irradiation by H^+^ leads to neutral or even enhancing recognition memory in doses of 0.5 or 1 Gy (0.2 or 0.5 keV/µm) [97,119]. More space-related combined irradiation (^12^C, 0.14 Gy, 10.3 keV/µm and 0.4 Gy γ-rays) did not affect the recognition memory [9] or even enhanced it (^12^C, 0.16 Gy, 10.3 keV/µm, and 0.3 Gy γ-rays) on the anxiolytic therapy background [82]. Mixed HZE irradiation at space-related dose 0.25 Gy did not affect the recognition memory, but impairment was observed at higher (0.5 or 2 Gy) doses [55]. Recently, the most progressive ground-based irradiation model (0.75 Gy; a combination of 33 nuclei, modeling the natural background in outer space) showed no abnormalities in the recognition memory [37]. The latest pool of data correlate with the neutral/positive results of H^+^ and HZE irradiation on the contextual and working memory described in Y-maze and PA tests. At the same time, the recognition memory impairments are often associated with high levels of anxiety.

### 5.5. Limitation of Some Cognitive Studies and Summarizing

We believe that in the tests with positive reinforcement or tests based on the instinctive behavior analysis (ORT, for example), the main cause is insufficient motivation. The lack of motivation to perform a number of tests in irradiated rodents has also been pointed out by other authors [114]. It also proves the absence of disturbances and even positive effects of IR (γ-rays, H^+^, HZE, H^+^ and γ-rays, HZE and γ-rays, mixed HZE) in space-related doses on memory or learning dynamics in tests using stressful conditions (MWM) or aversive stimuli as electric foot-shock (PA, Y-maze).

The violations of spatial memory retention (the probe test) in MWM are often considered as a separate indicator [109,120]. At the same time, it is inextricably linked to the dynamics of spatial learning and intended to confirm the result, indicating the possible cause of the violation. The probe test alone is hardly informative. Indeed, the conclusion about the spatial memory impairment based on the probe test with the unchanged positive dynamics of learning (on the level of naïve animals) seems imprudent.

In another test, the ORT, sometimes even naïve rodents are not interested in a new object at all, which does not allow to collect a sufficient sample size for analysis [12,74]. Our observations show that it is necessary to choose the objects which are attractive enough for animals and which could not serve as anything except the element of novelty. In ORT, only the first two–five minutes may be analyzed because that is the period during which the rats have been shown to be the most sensitive to novelty [121,122]. Specific behavior variables that may affect novel object recognition are not well studied, and there may be additional mechanisms that explain why an animal fails to recognize a familiar object or object location [123].

Earlier, we conjectured that the emotional state of rodents plays the primary role in the IR effect on cognitive abilities [54]. This hypothesis has recently been validated within the behavioral paradigms used [82]. A high level of anxiety can significantly affect the performance of ORT [124,125], but this is not an axiom [126]. At the same time, high anxiety rates can lead to improved learning dynamics in MWM by stimulating motivation in a stressful environment [54,127]. Indeed, mild stress can enhance learning and memory [128,129]. These data are also confirmed in a human study [130,131,132]. It probably matters in which paradigm the assessment of “state” anxiety occurs [81]. Another indicator of the emotional state—OEB—directly affects the performance in ORT and is regarded as a marker of undirected attention [133]. However, these characteristics of behavior can be considered as predictors of general learning ability only in combination with habituation [134]. Both OEB and habituation undergo significant modulation under the IR influence (see Section 4.3). Thus, radiation-induced rapid habituation may be responsible for a sharp decline in interest to new objects in ORT.

Summing up, IR exposure convenient to the interplanetary human mission (<2 Sv for rodents) either does not affect or even can enhance the performance in high-level cognitive tasks, and this effect becomes more pronounced when using animals with an evolutionarily more developed neocortex (non-human primates). This conclusion is also confirmed in a human study on the ISS. On the contrary, in low-level cognitive tasks, the effect is multidirectional; we also observe a high sensitivity to the emotional state, animal housing, and experimental conditions.

## 6. Direct and Indirect Effects of Irradiation: Nature of IR Positive Effect on Cognitive Abilities

IR effects can be divided into direct and indirect ones. The direct damaging effects of radiation include nuclear transmutations (neutron capture, inelastic scattering); secondary radiation generation (nuclear fragments, neutrons, electrons, etc.); production of free radicals and water radiolysis products (e^−^, OH, H_3_O^+^, H, OH^−^, H_2_, H_2_O_2_); and chemical bonds breaking [21,135]. Moreover, IR can influence the probability of quantum and reverse quantum tunneling [136]. A violation of the biomolecule’s chemical composition and structure, including the most dangerous multiple double DNA breaks, is the climax of the direct IR effect [137], which ends in cell death. If a particular threshold value of the absorbed dose is exceeded, the affected cell dies according to the necrotic scenario, no matter if it is a high-LET HZE particle or low-energy particle, which stopped in tissue—irradiation in Bragg peak [138]. Radiation damage of biological macromolecules is mainly mediated by the rupture of S–H, O–H, N–H, and C–H chemical bonds and occurs at the time interval 10^−12^–10^−14^ s. At 1 ps, irradiation damages C–C and C–N bonds on the background of beginning water radiolysis. At this time point, the indirect IR effects are manifested. The radical species recombination occurring one µs after the living matter irradiation ensures the homogeneous radiolytic medium formation within the local particle’s passage region, which makes this stage one of the most destructive [135]. For heavy particles, the diameter of such a region may exceed the neuron body size (50 μm) [139]. The secondary IR effects are very diverse and include oxidative stress [140], protein folding and endoplasmic reticulum stress [141], genetic mutations [142], inflammation [143], neurogenesis inhibition (see Section 8), impaired neuron and glial cell morphology [53,144], a wide range of molecular rearrangements [145,146], impaired synaptic transmission and neurotransmitters metabolism [83,147,148], and finally the apoptotic death of nerve tissue cells [146].

Living organisms and, in particular, its nervous tissue responds actively to the IR damage [149], initiating: DNA repair [150], antioxidant and anti-inflammatory mechanisms [151,152], neuro-regeneration, including neurogenesis activation [153,154], antiapoptotic mechanisms [155], molecular rearrangements of the neurotransmitter’s metabolism, and synaptic plasticity. Double DNA breaks and the subsequent activation of repair systems play an essential role in neural plasticity: both pathological and adaptive directions [156,157]. Some of these responses can be considered positively when estimating the CNS functions [9,14,39,54,158,159]. Indeed, a hyper-activation of some cellular processes prevents cognitive dysfunction [160]. At the same time, one hypothesis suggests that IR can cause a hyper-activation of recovery mechanisms in CNS that may temporarily lead to a seeming brain functions improvement but then followed by pronounced cognitive impairment [161].

CNS disinhibition by IR has been proposed as one of the mechanisms responsible for cognition enhancing [4]. The increasing 5-HT_2a_ content as well decreasing content of GABA in cortex may be a conductor of this phenomenon [9,39,162]. At the same time, HPA axis activation accompanies radiation [54], predetermining its stressful effect. The amygdala—the key structure in emotional information processing and implementation of emotionally-motivational behavioral forms—is also involved in nerve tissue response to irradiation [54,120,163]. In addition, the alterations in the expression of dopamine and serotonin transporters, catechol-O-methyltransferase, tyrosine hydroxylase, as well dopamine D_2_ receptor were found in various morphological structures of the brain [39,158]. The direction of these molecular rearrangements remains unclear. The hypothetical neurochemical and molecular nerve tissue rearrangements under the IR influence is presented in Figure 3.

Ultimately, the above effects lead to the alterations in CNS functions that we observe as motor, emotional, and cognitive changes. Conducting behavioral tests and molecular studies, a researcher observes not the primary effects of radiation (which pass very quickly), but the result of confrontation of two phenomena: the development and spread of the pathological process and the activation of reparative/compensatory function of the organism.

## 7. Combined Effects of Hypogravity and IR and Their Mechanisms

SFF have a combined effect on living organisms, essentially balancing each other. Data from ISS can serve as a source of such complex action [4,164]. The recent orbital “Twins” study (340 days on the ISS) showed that the cognitive abilities and the creative activity of the astronauts were significantly stimulated during the flight, but after returning to the Earth, this cognitive enhancement inverted [68]. No cognitive impairment was found in rats (nether males nor females) after the short-term orbital mission (16 days). However, the OEB change was noted; animals were also characterized by using other strategies to solve the tasks during the first few days of testing, but these differences normalized soon [165]. Some authors believe that the observed effects are more associated with the stress and hypogravity-induced impairment of coordination, including space motion sickness, than with the direct damaging effect of SFF [54,166,167].

More accurate data are reported by the ground-based experiments. The pre-irradiation by X-ray (0.05 or 0.15 Gy), followed by emotional stress (isolation), leads to the aggressive behavior suppression [168], which is typical for isolation as the only factor. Under the influence of the γ-rays the stress effect suppression was detected but only when these factors were applied simultaneously [169]. Relying on the interplanetary flights’ goal, the most interesting is the interaction between gravitational and radiation factors. It should be noted that the ground modeling of both hypogravity and IR is imperfect [4], and the results should be perceived accordingly. Antiorthostatic suspension (AS; hindlimb unloading, in other nomenclature) is the main ground modeling method of hypogravity effects in rodents. Even early studies of the IR and AS effects on the CNS functions demonstrated their opposability [170]. In another study, rats subjected to the combination IR (γ-quanta 0.5 Gy × 6, fractionated) and AS showed better learning tendency in differentiated motor-food conditioned reflex test [90]. In a similar experiment, lower γ-quanta doses (40 mGy, chronically 21 days) stimulated locomotor activity and risk appetite of mice previously exposed to AS [159]. Simultaneous AS (30 days) and irradiation (0.5 Gy × 6, fractionated) followed by irradiation by H^+^ (1.5 Gy, 0.4 keV/μm) results in the restoration of locomotor activity, ER, OEB, contextual memory, and have an anxiolytic effect in rats [39]. In a slightly modified experiment, a combination of AS (14 days) with 3 Gy γ-quanta (daily) followed irradiation by H^+^ (1.5 Gy, spread-out Bragg peak) showed a normalization of anxiety level, as well as the restoration of the contextual memory and somehow the improvement in MWM learning; however, in the latter case, the AS effect dominated [14]. Recently, using a more space-relevant irradiation model (mixed nucleus: H^+^, ^4^He, ^16^O, ^28^Si, ^56^Fe), the multidirectional interaction with AS (30 days with a break on day 5) was confirmed: the irradiation (1.5 Gy) blocked AS-induced spatial habituation learning impairment, whereas AS blocked the IR-induced (1.5 Gy, but not 0.75 Gy) depressive-like behavior [171]. Thus, we observe the surprise effect of IR on the background of the pathological process in the nerve tissue (caused primarily by stress and antiorthostatic suspension), which neutralize the negative effects on the CNS functions. The nature of this phenomenon is still unknown.

In response to the combined AS and IR action, the blood–brain barrier violation was detected, but no signs of the neuroinflammation were detected [172]. Remarkably, γ-rays (40 mGy, chronically) block the increase in a number of apoptotic cells caused by the AS action in nerve tissue [159]. This is in good agreement with the orbital experiment data (1 month, mice), which revealed an increase in antiapoptotic BCL-XL gene expression in the hippocampus, but in the hypothalamus and striatum, the opposite effect was found [155].

Until now, the precision neurochemical analysis was made only for the monoaminergic system of rodents subjected to the simultaneous action of SFF. Under the simultaneous action of IR and AS the serotonin metabolism in the prefrontal cortex, hippocampus, and striatum, as well as dopamine metabolism in the prefrontal cortex, are normalized [14,90]. At the same time, the increased noradrenergic prefrontal cortex innervation may be responsible for cognition enhancing [173].

The molecular changes caused by the AS and IR factor combination remain poorly understood. It was shown that the combined AS and IR action causes a significantly greater effect than the action of the individual factors and leads to the alteration of the pathways involved in neurogenesis and neuroplasticity, neuropeptides regulation, and cellular signaling [174]. Overexpression of serotonin transporter, catechol-O-methyltransferase in the hippocampus and striatum as well as tyrosine hydroxylase in the striatum was found under simultaneous IR and AS action [39].

## 8. IR Role in Neurogenesis: A Double-Edged Sword

The fact that the inhibition of neurogenesis is under the influence of various natures of IR is generally recognized. Neurogenesis is very sensitive to X-ray [175] and other IR. Several authors have shown that IR-induced impairment of neurogenesis is associated with cognitive disability [176,177]. However, the reverse data also exist [10,178,179]. Indeed, the hippocampal neurogenesis is not critically required for memory formation and learning [180]. Different authors still debate about how critical the neurogenesis preservation is for training in MWM [181,182]. At the same time, these disputes do not concern synaptogenesis [183]. Thus, the neurogenesis alterations (including IR-induced) cause learning disruption only in some behavioral paradigms. Additionally, they are associated with anxiety levels (see Section 4.1), which may affect learning outcomes in some tasks.

For X-rays (150 keV) and γ-rays, the acute dose causing neurogenesis inhibition is ~1 Sv [175,184] and ~6 Sv/year in chronic experiment, respectively [151]. Irradiation in mixed fields (0.34 Gy neutron 0.5–30 MeV and 0.36 Gy γ-rays) also demonstrates impaired neurogenesis without cognitive deficits [178]. A number of studies have shown the inhibition of neurogenesis by H^+^ and HZE irradiation with a wide value of Z and energies: H^+^ (0.5 Gy, 0.2 keV/μm), ^12^C (1 Gy, 8 keV/μm), ^28^Si (0.2 Gy, 67 keV/μm, male mice), or ^56^Fe (1 Gy, 175 keV/μm) [48,59,185,186]. In contrast, no neurogenesis abnormalities were revealed upon 0.2 Gy ^28^Si (67 keV/μm, female mice) or 0.5 Gy ^56^Fe (175 keV/μm) exposure [48,59,104]. These data suggest that there exists a threshold of dentate gyrus cells penetration (~200 particles/cm^2^), which is necessary to initiate disturbances (Figure 4). However, this speculative assumption requires further statistical approval and careful validation.

An important ground modeling experiment characteristic is the dose absorbance method: acute, chronic, or fractionated approach. Chronic HZE irradiation modeling is quite difficult, so there is a number of methods to simulate it: the use of γ-rays or neutron sources and/or acute or fractionated HZE irradiation. Fractionated γ-irradiation (^137^Cs 1 Gy/min source, 0.1 Gy and 2 Gy after 24 h) of mice significantly reduces the formation and maturation of new neurons compared to 2 Gy acute irradiation [187]. In contrast, more pronounced fractional (1 Gy × 5) γ-irradiation significantly reduces the negative effect of the γ-rays on neurogenesis compared to the acute irradiation [188]. Finally, the ultra-low doses of chronic γ-ray radiation (1 mSv/day or 20 mSv/day 300 days) show no influence on neurogenesis. Moreover, the signs of an anti-inflammatory effect and a decrease in lipid peroxidation were detected but only at a total dose of 0.3 Gy in mice with *APOE* gene expression inactivation [151]. The research using HZE showed a moderate decrease (after 24 h) in the negative effect of radiation on the proliferating hippocampal cell number when the dose of ^56^Fe (238.8 keV/μm) was fractionated (0.2 Gy × 5, daily, 0.2 Gy/min) compared to the acute irradiation (1 Gy, 1 Gy/min) [38]. Thus, the particular scenario implementation depends on many parameters, where dose rate and fluence (for particles) are the main predictors. In general, these data seem to be positive for astronauts, since they indicate a significant reduction in risk when fractionating a large dose over time. The recent research findings show the stunning regenerative and compensatory capabilities of the brain, which can cope with radiation exposure beneath a certain threshold.

Can nerve tissue regenerate after radiation damage? Indeed, the proliferative response after the cell’s depletion via IR-induced apoptosis may represent the recruitment of relatively quiescent stem/precursor cells [175,189]. Moreover, local irradiation (10 Gy, X-rays) of the right subventricular zone enhances the proliferation rate and neuro-reparation in response to chemically induced demyelination in the striatum [153]. It has been hypothesized that the nerve stem cells are relatively radioresistant and can be recruited by the IR-induced apoptosis [190]. This scenario is true, at least for the radiation damage in doses up to 5 Gy for X-rays [191,192] and, more importantly, in doses up to 1 Gy ^12^C nuclei (whole-body, 8 keV/μm), 0.3 Gy ^56^Fe (147 keV/μm), and perhaps even 0.5 Gy ^56^Fe (175 keV/μm) [59,184,185]. The IR-induced increase in the number of apoptotic cells can be maintained for 9 months and, therefore, continuously stimulate neurogenesis [154]. This conclusion was confirmed by the preservation of the hippocampal neurons number 12 months after brain radiation damage at the dose of 45 Gy (γ-rays, fractionated 5 Gy × 9, 661.7 keV) [193]. Importantly, the stimulation of neurogenesis via IR-induced apoptotic nerve cell’s death can be used as a therapeutic approach, for example, in the terminal stages of neurodegenerative diseases, when the expected positive effects obviously outweigh the possible delayed negative effects of radiation.

Radiation can suppress neurogenesis via various mechanisms; however, the inflammation is considered as one of the main causes [59,194,195]. To date, the study of radiation-induced neuroinflammation at space-related doses has received insufficient attention. It is known that irradiation with higher doses of γ-rays (10 Gy) leads to activation of glia and increases the content of pro-inflammatory factors, such as IL-1, TNF-α, INF-γ, and IL-6 [194]. Irradiation with ^56^Fe (175 keV/μm; 1, 2 or 4, but not 0.5 Gy) leads to microglia activation (GFAP and CD68 as markers) [59]. The increased content of pro-inflammatory C-C chemokine receptor type 2 was detected in response to irradiation by ^12^C (13 keV/μm) and ^56^Fe (147 keV/μm) nuclei at absorbed doses of 1, 2, or 3 Gy at 30 days after irradiation [196]. The only study with space-relevant doses relying on the microglia activation marker CD68 revealed inflammation in the medial prefrontal cortex of mice after irradiation by ^16^O (15.8 keV/μm) or ^48^Ti (106 keV/µm) at doses of 50 and 300 mGy [58]. Little is known about the anti-inflammatory effect of IR, which was shown after H^+^ (0.1–2 Gy, 0.2 keV/μm) irradiation with intercellular adhesion factor 1 as a marker [186]. The molecular mechanism of radiation-induced neuroinflammation and its proliferation is still unresolved, but a recent study showed that direct interaction of microRNA-181b-2-3p with SRY-box transcription factor 21 might be crucial link in radiation-induced microglial activation and proliferation [197]. At the same time, the effectiveness of the antioxidant in IR-induced neurogenesis impairment restoration indicates the oxidative stress is the main cause of inflammation [198].

Of particular interest are the HZE effects on the expression of neurotrophins. The orbital experiment (~1 mSv/day × 30 days; together with hypogravity factor) with mice showed no changes in the expression level of the brain-derived neurotrophic factor (BDNF) and the TrkB receptor [155]. A decrease in the BDNF content was found after the mixed HZE irradiation at an absorbed dose of 2 Gy (male only) but not space-related 0.25 or 0.5 Gy. Moreover, the female mice subjected to the same dose formed a tendency to increase the BDNF content in the neocortex [55]. At the same time, another study with combined irradiation (H^+^ 0.24 keV/µm, ^4^He 1.6 keV/µm, ^16^O 25 keV/µm, ^28^Si 78 keV/µm, ^48^Ti 107 keV/mm, and ^56^Fe 151 keV/mm in proportion of adsorbed dose 50:20:7.5:7.5:7.5:7.5, respectively; 0.5 Gy totally) found increased BDNF content in neocortex in male but not female mice [199]. Future studies are designed to confirm or refute this assumption.

## 9. Neurodegenerative Processes in Light of Radiation

A progressive neurodegenerative process that could be, hypothetically, initiated by IR during the interplanetary flight would be the most unfavorable scenario for the future space colonists. Taking this into account, the researchers could not ignore the effect of HZE on the initiation and progression of proteinopathies, a class of neurodegenerative diseases characterized by the accumulation of aggregated proteins in nerve tissue cells or in extracellular space [200].

The data on the effect of radiation on the neurodegenerative process are summarized in Table 2. The acute X-ray irradiation (100 mGy, 200 keV) does not affect the pathological aggregation of amyloid-beta (Aβ) and tau protein, the main components of protein aggregates in Alzheimer’s disease (AD) and Parkinson’s disease [201,202]. It should be noted that chronic exposure (1 mGy/day, 300 days) suppresses neuroinflammation in apolipoprotein E knock-out mice—a model of atherosclerosis and AD [151]. The first studies of the HZE effects were not so encouraging. The irradiation with 0.1 or 1 Gy ^56^Fe (147 keV/μm) was shown to increase Aβ plaque pathology in an APP/PS1 mouse model of AD expressing human transgenes for amyloid precursor protein (APP) and presenilin-1; both mutations are under the control of the Thy1 promoter [203]. It is noteworthy that this effect was found only in males but not in females (1 Gy), and the equivalent dose used in this experiment significantly exceeds (~2 times for 100 mGy) the predicted radiation load during the 860-day Martian mission. In another study, the positive effect of H^+^ radiation (1, but not 0.1 or 0.5 Gy, 0.57 keV/µm) on the number of Aβ plaque in the neocortex but not the hippocampus of male APP/PS1 transgenic mice was revealed. At the same time, the irradiation did not affect the cognition in behavior tests and expression of pro-inflammatory cytokines or presynaptic protein synaptophysin in mice cortex, which, according to the authors, argues against IR and AD additive effects [204]. The further increase of LET, namely irradiation by ^12^C (15 keV/µm) at doses of 50 or 100 mGy, did not result in significant accumulation of APP, Aβ, tau, and phospho-tau in hippocampal CA1 regions [49]. The irradiation by 0.1 or 0.5 Gy ^56^Fe (147 keV/µm) led to a decrease in both cerebral Aβ levels and microglia activation but only in female transgenic AD modeling mice (APPswe/PS1dE9) without affecting naïve animals [16]. Finally, combined irradiation (^12^C 0.18 Gy, 10.3 keV/μm and γ-rays 0.24 Gy) had an anxiolytic effect and stimulated OEB in the mice with tauopathy (Tau P301S line) and also improved odor discrimination in the mice with cerebral amyloidosis (5xFAD line). There was no effect on learning in MWM and the contextual memory in PA [15]. Thus, a nootropic effect against the background of a neurodegenerative process has been discovered.

Importantly, the number of mature fibrils is not a necessary and sufficient marker for assessing the neurodegenerative process. It is generally accepted that intermediate forms of protein aggregation (protofibrils and oligofibrils) are the main neurotoxic component, while mature fibrils are even considered as a protective cellular mechanism for the utilization of toxic forms of aggregating protein [205]. Moreover, an important role in the proteinopathies pathology is played by the depletion of the normal protein function. While protein is “locked” in the aggregates, there is a loss of functionally active protein [206]. Until now, there is no study of HZE influence on the content of the protofibrils and oligofibrils of pathologically aggregating protein. In any case, the Aβ plaque decrease in the brain tissues did not show any cognitive benefit and cannot be considered as a positive signal in the treatment of the neurodegenerative disease [207,208].

Interestingly, the combined irradiation (0.14 Gy ^12^C and 0.4 Gy γ-rays) causes a GABAergic control weakening of the cortical glutamatergic neuronal networks [9]. Indeed, a decrease in cortical GABA and subsequent excitation is typical for traumatic brain injury (TBI) [209]. At the same time, TBI is a risk factor for AD [210]. However, irradiated animals did not show any other signs of TBI-like pathophysiology [9,83], and the behavioral analysis has not revealed traumatic encephalopathy signs in rats during 7 months after irradiation [9].

To date, chronic low-dose γ-radiation is considered as a physiotherapeutic approach in AD treatment [211,212]. At the same time, the data on HZE-irradiation are insufficient for an unambiguous conclusion. Nevertheless, the summation of all literature data allows us to believe that the space-related (<2 Sv) doses of HZE (including such high-LET particles as ^56^Fe nuclei) will have no neurodegenerative-like effect. And more importantly, in some doses, the HZE irradiation has a nootropic effect on the rodents with neurodegenerative processes. The future studies focusing on low-dose chronic HZE exposure will allow to bring finality to this controversial issue.

## 10. Conclusions

To form the conclusions, we should place the data in priority order: firstly, those obtained during orbital missions (considering the complex impact of space flight factors), then ground-based modeling experiments using mixed IR (including HZE and/or pre-irradiation by low-LET IR) in space-related doses (chronic, prolonged, or fractionated impacts, first), and after that—all the others. With regard to living organisms, the priority was given to the data from humans, after that from non-human primates, and finally from rodents. This is based on the following reasoning. On the one hand, we aim to choose the most relevant irradiation model of the exposure in interplanetary space or on the planet surface (chronic, multicomponent with a predominance of low-LET component). On the other hand, we need the results close to the human level, which are the most relevant in primates modeling. 

Thus, natural IR (orbital experiments, including ISS) is safe for animal’s cognitive abilities and human intelligence/creativity. It is noteworthy that this conclusion is confirmed in ground-based experiments in rodents, including experiments with space-related HZE compositions and doses. Moreover, in the conditions of a high-level cognitive task, the inexplicable enhanced abilities are observed.

Another important issue is the impact of IR on the emotional state (anxiety, ER, OEB, habituation) and depressive-like behavior. The alterations of the emotional state (anxiety, firstly) underlie one of the hypotheses that explain cognitive impairment, as well as the irreproducibility of the negative IR effect on performance in low-level cognitive tasks. At the same time, it is necessary to be careful when transferring data on the IR effect on the emotional state from rodents to humans.

The mechanisms behind the inexplicable positive effects of HZE irradiation (pro-cognitive, nootropic, etc.) remain unsolved. Hyperactivation of reparative and neurocompensative mechanisms at molecular, cellular, and tissue levels can underlie these effects. The CNS disinhibition and the neurogenesis stimulation at the background of neurons and other cells death (including cells with an existing pathology) are the key characteristics of these processes. A number of biomolecules have been proposed as a conductor for the positive effects of IR, but there is no sufficient evidence to justify their role. The molecular mechanisms of cognitive enhancing effects require careful study in the future.

We believe that the future studies of the IR effects on the CNS functions should include an extended number of behavioral tests to assess emotional state and cognitive abilities focusing on the high-level cognition tasks for rodents. The analysis of the emotional state should be preferably accompanied by the assessment of habituation, OEB, and ER. Thus, the data on rodents will remain relevant, but nevertheless, the preference should be given to primates, where it is possible to fully evaluate the operator’s activity in working with a computer (fine motor skills when working with a joystick, tracking tasks, switching attention, decision-making), including virtual reality.

A number of already obtained results require the special attention of researchers. Firstly, the scientists describe higher resistance to HZE in female rodents, so sex-dependent radiosensitivity is to be researched. Secondly, the positive effect of IR and HZE, in particular, manifested both as an independent phenomenon and against pathological background (in this case the IR effect is not detected for naïve animals). Thus, a number of studies have shown that the radiation normalizes the emotional state and cognitive abilities affected in rodents subjected to AS—a model of hypogravity. The mechanism of this phenomenon remains undisclosed, but verily, it is an intriguing discovery that requires careful study. Moreover, the recent evidence suggests that hypogravity may have an anticancer effect [213]. Thus, this “neutralizing” effect can be mutual: IR and hypogravity can block each other’s negative effects during the space flight. This state of affairs might be extremely speculative; however, if this hypothesis is confirmed, it will definitely change our understanding of the SFF effects and the space environment in general.

## Figures and Tables

**Figure 1 biology-12-00400-f001:**
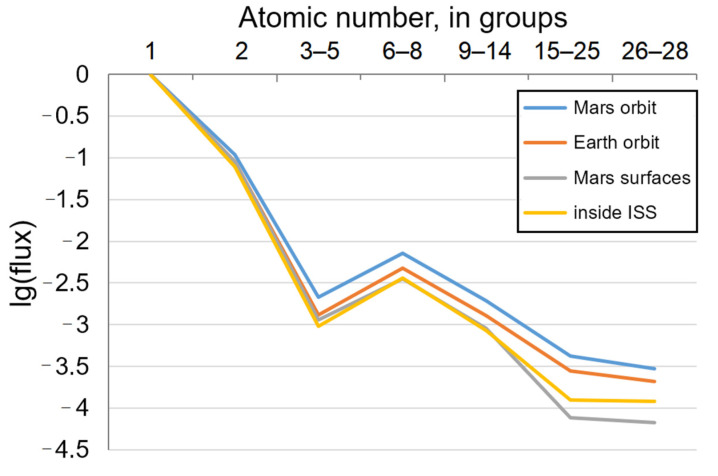
The flux of heavy-charged high-energy particles observed at several points in space: Mars orbit, Earth orbit, Mars surface, and inside the International Space Station (ISS). The data are presented as the decimal logarithm of the particle flux (ordered by groups by atomic number) relative to the proton flux taken as 1. Primary data taken from the following sources [29,30,31,32].

**Figure 2 biology-12-00400-f002:**
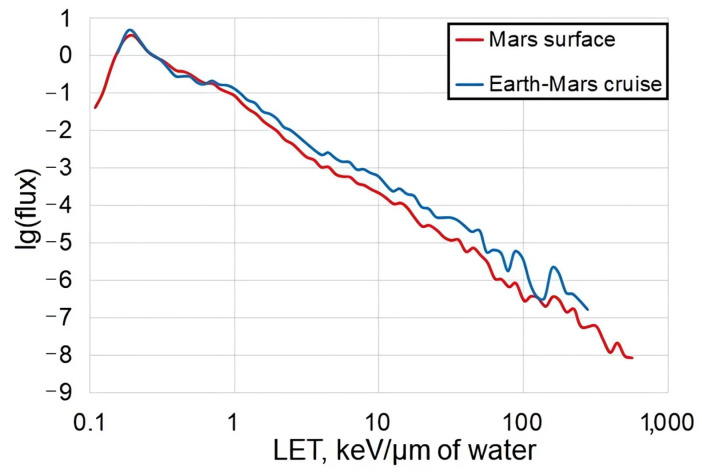
Comparison of charged particle LET spectrum measured on the Mars surface (red) to that measured during Earth–Mars cruise inside the MSL spacecraft (blue) with variable shielding. The data are presented as the decimal logarithm of the particle flux. Logarithmic scale on *X* axis was applied. LET—linear energy transfer. Primary data taken from [33].

**Figure 3 biology-12-00400-f003:**
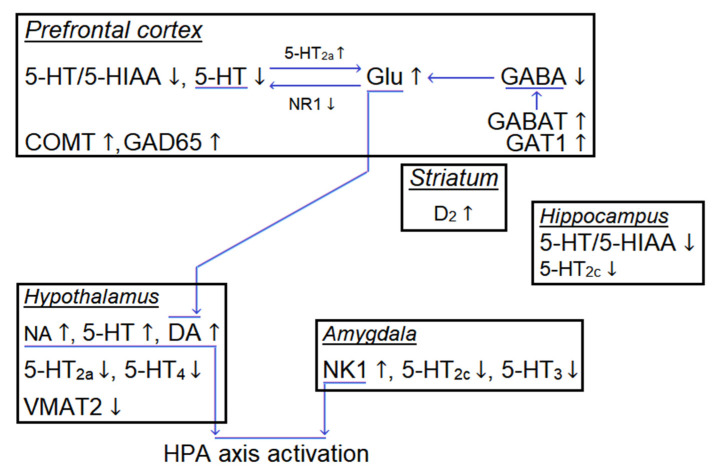
The hypothetical model of nerve tissue response within the boundaries of the studied neurotransmitter systems to the ionizing radiation exposure. Morphological structures with identified changes are circled by rectangles. Vertical arrows near enzymes, transporters, and receptors indicate a change in the level of mRNA expression (up—increase, down—decrease) under the radiation exposure; arrows near neurotransmitters indicate a change in total concentration. Long blue horizontal and vertical arrows indicate the stimulating effect of one neurotransmitter system on another. If there is an assumption of the conductor of this effect, it is indicated above or below the arrow. The acting and target neurotransmitter system are underlined by a blue line. DA—dopamine, NA—noradrenalin, 5-HT—serotonin, 5-HIAA—5-hydroxyindoleacetic acid, Glu—glutamate, GABA—γ-aminobutyric acid, COMT—catechol-O-methyl transferase, GABAT—γ-aminobutyric acid transaminase, GAD65—glutamate decarboxylase 65 kDa, VMAT 2—vesicular monoamine transporter 2, NK1—neurokinin-1 receptor, HPA—hypothalamic-pituitary-adrenal axis. Data is integrated from the following sources [9,14,39,83,158].

**Figure 4 biology-12-00400-f004:**
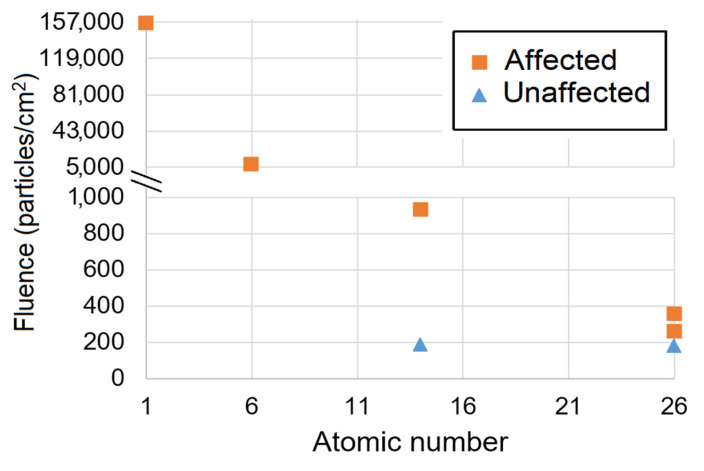
Effect of heavy charged particle’s fluence on adult neurogenesis. The graph shows cases of impaired neurogenesis (marked as “Affected”), as well as the lack of influence on neurogenesis (marked as “Unaffected”).

**Table 1 biology-12-00400-t001:** Anxiety and ionizing radiation.

Object	Dose and Composition (Age of Irradiation)	Condition of Irradiation	Anxious Behavior (Age of Testing)	Reference
Sprague-Dawley male rats	γ-rays, 661.7 keV, 0.5 or 1, or 2, or 4 Gy; ^4^He, 0.9 keV/µm, 1 or 5, or 10, or 50, or 100 mGy (~45–50 days, relying on [77])	Acute	Increased except 4 Gy γ-rays (2.5–6.5 months)	[74]
C57BL/6 male mice	H^+^, 0.5 keV/µm, 0.5 or 1 Gy (6 months)	Acute	Increased (15 months)	[75]
Wistar male rats	H^+^, 0.4 keV/µm, 1.5 Gy combined with γ-rays, 3 Gy (3 months)	γ-rays fractionated × 6; nuclei—acute	Not changed (5 months)	[39]
Wistar male rats	^12^C, 0.4 keV/µm, 0.14 Gy combined with γ-rays, 661.7 keV, 0.4 Gy (3 months)	Acute	Increased (3 or/and 11 months)	[9,54]
C57BL/6J male mice	^28^Si, 67 keV/µm, 0.2 or 1 Gy (~70 days)	Acute	Increased only at 1 Gy (4 months)	[48]
Thy1-EGFP male mice	^48^Ti, 126 keV/µm, 0.3 Gy (6 months)	Acute	Increased (~12 months)	[58]
Fischer 344 male rats	^48^Ti, 134 keV/µm, 10 or 100 mGy (15 months); ^16^O, 14.2 keV/µm, 1 mGy; ^4^He, 0.9 keV/µm, 0.1 or 0.5, or 1 mGy (15 months)	Acute	Increased only when irradiated by ^48^Ti or ^16^O (15 months)	[76]
Fischer 344 male rats	^48^Ti, 134 keV/µm, 10 or 100 mGy (2 or 11 months); ^16^O, 14.2 keV/µm, 1 mGy; ^4^He, 0.9 keV/µm, 0.1 or 0.5, or 1 mGy (2 or 11 months)	Acute	Not changed (2 or 11 months)	
Sprague-Dawley male rats	^56^Fe, 147 keV/µm, 1.5 Gy (3 months)	Acute	Not changed (6 months)	[73]
C57BL/6J male mice	mixed 33 beams of nuclei (H^+^, ^4^He, ^12^C, ^16^O, ^28^Si, ^48^Ti, ^56^Fe) with different energy, 0.75 Gy totally (6 months)	Acute	Not changed (10.5 months)	[37]
C3H male mice and BALB/c female mice	^252^Cf source of neutrons and γ-rays, 0.12 or 0.2, or 0.4 Gy (2 months)	Chronic, 400 days	Increased only at 0.4 Gy, C3H mice (20 and 23 month); not changed in other	[12]

**Table 2 biology-12-00400-t002:** Neurodegeneration and ionizing radiation. MWM—Morris water maze; APP—amyloid precursor protein; Aβ—β-amyloid peptide; tau and phospho-tau—tau protein and its phosphorylated form, respectively.

Object	Dose and Composition	Behavior	Protein Aggregation	Reference
C57BL/6J Jms mice	100 mGy X-ray, 200 keV	MWM not changed.	Aβ, tau, and phospho-tau plaque in hippocampus not changed.	[201]
APP/PS1 mice	^56^Fe, 147 keV/μm, 0.1 (male and female) or 1 Gy (only female)	Impairment of contextual fear conditioning (only 1 Gy, male); recognition memory decreased.	Increased Aβ plaque.	[203]
APP/PS1 mice	H^+^, 0.57 keV/µm, 1 Gy, but not 0.1 or 0.5 Gy	MWM and Barnes maze not changed.	Increased Aβ plaque in the neocortex, but not hippocampus.	[204]
C57BL/6J Jms mice	^12^C, 15 keV/µm, 50 or 100 mGy	MWM not changed.	APP, Aβ, tau, and phospho-tau plaque in hippocampal CA1 regions not changed.	[49]
APP/PS1 mice, C57BL/6J mice	^56^Fe, 147 keV/µm, 0.1 or 0.5 Gy	Improved motor learning (0.5 Gy, female APP/PS1); reduced grip strength (female APP/PS1); spatial memory in the Y maze not changed.	Decreased Aβ plaque (APP/PS1); not changed (C57BL/6J).	[16]
Tau P301S mice 5xFAD mice	^12^C, 10.3 keV/μm, 0.18 Gy combined with γ-rays, 0.24 Gy	Anxiolytic effect and stimulated orientation and exploratory behavior (Tau P301S); improved odor discrimination (5xFAD).	Not studied.	[15]

## Data Availability

No new data were created or analyzed in this study. Data sharing is not applicable to this article.

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
