# Peer review of "The Effects of Galactic Cosmic Rays on the Central Nervous System: From Negative to Unexpectedly Positive Effects That Astronauts May Encounter"

_biology, 2023, doi:10.3390/biology12030400_

Round 1
Reviewer 1 Report
This is an interesting study. The following issues should be addressed.
1. “led to violations in a number of cognition tasks”. Please rephrase and replace the word violations.
2. This manuscript is very hard to read. Large blocks of texts. Summarizing tables would help a lot here.
3. Ground-based studies involving hindlimb unloading do not seem included.
4. Studies involving omics data following irradiation do not seem involved.
5. Effects of irradiation on sleep and circadian rhythms do not seem included.
6. Effects of irradiation on ephys measures do not seem indicated.
7. DNA DSBs are shown to be induced during normal learning and memory. In the review, only detrimental effects of DSBs are indicated.
8. The author are encouraged to include one or more schematic figures summarizing the main findings.
9. Effects of irradiation on microglial activation and neuroinflammation in general does not seem considered.
Author Response
This is an interesting study. The following issues should be addressed.
- “led to violations in a number of cognition tasks”. Please rephrase and replace the word violations.
In the revised version of the manuscript, we corrected this phrase
- This manuscript is very hard to read. Large blocks of texts. Summarizing tables would help a lot here.
In the revised version of the manuscript, we added two tables to sections "4.1. Anxiety and ionizing radiation" (Line 246) and "9. Neurodegenerative processes in light of radiation (Line 777). At the same time, the section "5. Cognition and ionizing radiation" has been fragmented to present the data better. As results of cognitive researches are not always unambiguous, different tests are applied and authors specify a number of nuances in the received data (it was not possible to collect representative groups or the part of animals appeared radio-resistant, for example), we did not find possibility for this section to present the table, it would appear too cumbersome.
- Ground-based studies involving hindlimb unloading do not seem included.
We review a number of works using the hindlimb unloading technique in section "7. Combined effects of hypogravity and IR and their mechanisms" (Lines 604-624). However, since the review is devoted to the effects of ionizing radiation, we have considered only those works in which these two factors act in combination.
- Studies involving omics data following irradiation do not seem involved.
In the manuscript, we cited several works devoted to transcriptome analysis (Line 540). However, despite the discovery of many genes whose expression was altered after irradiation, there is almost no pharmacological validation of these results. To date, it is impossible to speak precisely about the role of particular changes at the genome or proteome level in CNS radiation damage. The presented review is mainly devoted to functional (behavioral) changes in the organism under the influence of ionizing radiation. Adding these molecular studies will lead to an extreme increase in the length of the text and will exceed the limit of a "good" review.
- Effects of irradiation on sleep and circadian rhythms do not seem included.
Indeed, we have not considered these works. Most of the data on the effect of spaceflight factors on sleep and circadian rhythms were obtained on the ISS, and it is almost impossible to isolate the effect of each of the factors (ionizing radiation, hypogravity...). This is a very extensive topic and its consideration would extremely increase the volume of the manuscript. Unfortunately, the coverage of these effects is beyond the goals of our review.
- Effects of irradiation on ephys measures do not seem indicated.
In the revised version of the manuscript, we added the required information (Lines 154-172)
- DNA DSBs are shown to be induced during normal learning and memory. In the review, only detrimental effects of DSBs are indicated.
Thank you for the very valuable comment. This is another of the understudied effects of radiation exposure that may be responsible for the pro-cognitive effects of radiation exposure. In the revised version of the manuscript, we have added this information (Lines 546-548)
- The author are encouraged to include one or more schematic figures summarizing the main findings.
Indeed, adding a schematic figure that would summarize the main findings would be very useful. However, in the review we show the multidirectional effects of irradiation rather than draw any unambiguous conclusion. In fact, we have no reason to state definitely, for example, the anxiogenic or procognitive effects of irradiation... A particular result manifests itself only in a certain dose and scheme of irradiation, and the results of several works may contradict each other. The purpose of our work was to show this phenomenon of "multidirectionality" and to shift the emphasis from the unambiguous "negative influence on CNS functions" to "multidirectional influence including positive effects". Unfortunately, we did not find it possible to present a simple (without many refinements and limitations) schematic figure that would illustrate the main findings. In doing so, we shortened the section "10. Conclusions" for a better and more concise (easy to understand) presentation of the main conclusions of the work.
- Effects of irradiation on microglial activation and neuroinflammation in general does not seem considered.
In the revised version of the manuscript, we added the required information (Lines 711-725)
Reviewer 2 Report
GCR exposure causes multidirectional effects on cognition, which may be associated with emotional state alterations. The review summarized the relationship between the emotion, cognition, and IR. However, there are some problems to be solved.
1. The Section Conclusions is too complicated. One paragraph is enough. The other contents may be supplemented in the other parts.
2. There are too many grammar errors. The writers should revise carefully. For example, “However, γ-irradiation alone in the dose range of 0.5–2 188 Gy leads to anxiety increase“.
3. Cognition includes many abilities, such as memory, decision-making, learning, etc. It is suggested Section “5. Cognition and ionizing radiation” be classified into the detailed abilities.
4. How can we simulate the space environment on the ground?
It is suggested major revisions.
Author Response
GCR exposure causes multidirectional effects on cognition, which may be associated with emotional state alterations. The review summarized the relationship between the emotion, cognition, and IR. However, there are some problems to be solved.
- The Section Conclusions is too complicated. One paragraph is enough. The other contents may be supplemented in the other parts.
In the revised version of the manuscript, we shortened the section "10. Conclusions" for a better and more concise presentation of main findings
- There are too many grammar errors. The writers should revise carefully. For example, “However, γ-irradiation alone in the dose range of 0.5–2 188 Gy leads to anxiety increase“.
We have carefully checked the text in the revised version of the manuscript.
- Cognition includes many abilities, such as memory, decision-making, learning, etc. It is suggested Section “5. Cognition and ionizing radiation” be classified into the detailed abilities.
In the revised version of the manuscript, the section "5. Cognition and ionizing radiation" has been fragmented to present the data better.
- How can we simulate the space environment on the ground?
In the revised version of the manuscript, we added the required information (Lines 134-152)
Round 2
Reviewer 1 Report
The authors did a reasonable job addressing the raised concerns. It is a pity that many comments did not result in adding the related information into the manuscript and that a summarizing figure was not include as it would definitely strengthen this review.
Reviewer 2 Report
I suggest acceptance.